# INTERFAIR: Debiasing with Natural Language Feedback for Fair Interpretable Predictions

**Bodhisattwa Prasad Majumder**[*1], **Zexue He**[*2], and **Julian McAuley**[2]
[1]Allen Institute for AI
[2]University of San Diego
bodhisattwam@allenai.org, {zehe, jmcauley}@ucsd.edu
*contributes equally

## Abstract

Debiasing methods in NLP models traditionally focus on isolating information related to a sensitive attribute (e.g. gender or race). We instead argue that a favorable debiasing method should use sensitive information 'fairly,' with explanations, rather than blindly eliminating it. This fair balance is often subjective and can be challenging to achieve algorithmically. We explore two interactive setups with a *frozen* predictive model and show that users able to provide feedback can achieve a better and *fairer* balance between task performance and bias mitigation. In one setup, users, by interacting with test examples, further decreased bias in the explanations (5-8%) while maintaining the same prediction accuracy. In the other setup, human feedback was able to disentangle associated bias and predictive information from the input leading to superior bias mitigation and improved task performance (4-5%) simultaneously.

## 1 Introduction

Debiasing human written text is an important scientific and social problem that has been investigated by several recent works (Zhang et al., 2018; Jentzsch et al., 2019; Badjatiya et al., 2019; Heindorf et al., 2019; Ravfogel et al., 2020; Gonen and Goldberg, 2019; He et al., 2021). These methods primarily try to eliminate the biased information from the model's internal representations or from the input itself, disregarding the task performance during the process. However, in an ideal situation, a model should use only the necessary amount of information, irrespective of bias, to achieve an acceptable task performance. This trade-off between task performance and bias mitigation is subjective or varies between users (Yaghini et al., 2021) and is often hard to achieve via learning from data (Zhang et al., 2018; He et al., 2022). Figure 1 shows the limit of an algorithmic approach where ignoring all gendered information can lead to a wrong result.

However, a user can potentially further tune the model's belief on the bias, leading to a correct prediction while minimally using biased information. While interactive NLP models recently focused on model debugging (Tandon et al., 2021, 2022), improving explainability in QA (Li et al., 2022b), machine teaching (Dalvi et al., 2022), critiquing for personalization (Li et al., 2022a), and dialog as a more expressive form of explanations (Lakkaraju et al., 2022; Slack et al., 2022), we focus on an under-explored paradigm of model debiasing using user interactions. Objectively, we allow users to adjust prediction rationales at the test time to decrease bias in them, addressing the subjective aspect of fair and transparent debiasing.

In this paper, we propose INTERFAIR, a modular interactive framework that (1) enables users to provide natural language feedback at test time to balance between task performance and bias mitigation, (2) provides explanations of how a particular input token contributes to the task performance and exposing bias, and finally (3) achieves better performance than a trained model on full-text input when augmented with feedback obtained via interactions.

## 2 Background: Debiasing with Rationales

An interpretable debiasing algorithm produces a rationale along with a prediction of the original task to expose the amount of bias or sensitive information used. Precisely, a rationale is the minimal and sufficient part of the input responsible for the prediction. For text input, let the predictive input tokens for the task output be called *task rationales* and tokens revealing sensitive information be called *bias rationales*. Since the model solely uses the rationales to predict the task output, these rationales are highly faithful (Jain et al., 2020).

According to He et al. (2022), it is possible to attach an importance score for each token to be included in the respective task or bias rationales. Traditional debiasing algorithms face two failure

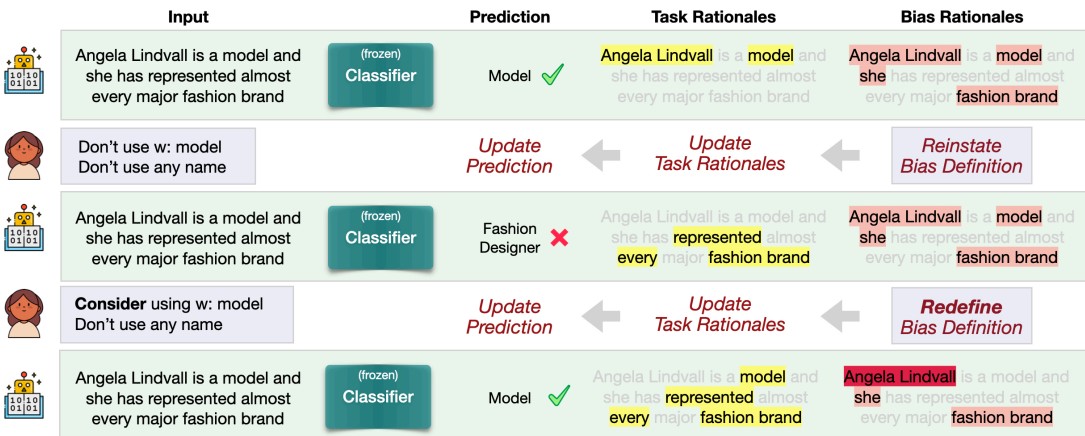

Figure 1: **Pipeline**. An algorithmically debiased model still suffers from generating biased rationales. Users interact with the final model states and perturb them using language feedback further to decrease bias and/or improve task performance.

modes: 1) it produces a correct task prediction but with a *highly biased rationale* and 2) it produces a *wrong task prediction* but a rationale with low bias. He et al. (2022) argue that weighing less on high-bias and high-task important tokens and promoting their low-bias replacements can simultaneously address both of the failure modes. However, this technique is not perfect, and Figure 1 shows a limiting case of this approach that opens up further room for improvement.

## 3   INTERFAIR: Interactive Fair Debiasing

We highlight that even an algorithmically debiased model can have failure modes and one potential option is to *fix* the problem at the inference time. We argue that human users are better at fixing the failure cases that a model is unable to learn from the training data. We also assume that the model parameters remain frozen during the fixing process, and users only interact with the final prediction and its associated hidden model states.

**Task and Base Model**   We start with a frozen model that is algorithmically debiased and allow users to interact and edit its rationale at the inference time towards lower bias. Since rationales are tied to task prediction, the user should edit them without lowering the task performance. Primarily, the users are encouraged to find better low-bias replacements for tokens highly important for both task performance and revealing bias. To this end, we hypothesize a system, **INTERFAIR**, to achieve a fair balance between task performance and bias.

For the scope of this paper, we use classification as the predictive task and text only as the input modality. For the base model, we use an LSTM classification model, trained using the procedure described in He et al. (2022). The classification model generates a prediction and a pair of normalized scores (between 0 and 1) for each input token for its contribution toward task rationale and bias rationale. While large language models (LLMs) can be superior classifiers, the opaqueness of these models hinders faithful perturbation of rationales, which is one of the goals of this work.

During operation, the user queries with a text input for the classification task (e.g., predicting the profession from a biography) and a known bias variable (e.g., gender). After querying, the user receives the prediction, rationales (with importance scores) for the task prediction, and the bias variable. Since the goal is to potentially disentangle the bias from the predictive task, we restrict users to directly modify the bias rationales only. A change in the bias rationales will trigger a change in the task rationales and, finally, in the prediction. Since rationales are in natural language (tokens), we enable users to interact in natural language (NL). INTER-FAIR converts the NL feedback to be actionable for the model to update its rationales.

### 3.1   Parsing NL Feedback

Rationales are presented to the users with importance scores for each input token (see Figure 1). To directly modify the bias rationales, users can increase or decrease the bias importance score for each token accordingly. For example, in the Figure 1 example, it is prudent to decrease the bias importance for the word `model` and increase the bias importance for `Agnela Lindvall`.

The simplest form of feedback is to provide feedback on the bias importance of a certain input token

| Example | $k$-shot | IID | Comp | Overall |
|---|---|---|---|---|
| [Input] Angela Lindvall is a model and she represented (...) [Bias] Gender | Model: GPT-3 (text-davinci-003) | | | |
| [Feedback] Angela Lindvall is a woman's name [Parse] High, High, NA, NA, NA, NA, NA, NA (...) | 5 shot | 58.7 | 34.2 | 46.5 |
| | 10 shot | 74.2 | 45.8 | 60.0 |
| | 20 shot | **83.8** | **60.1** | **71.9** |

Table 1: **NL feedback parser.** Parse example; parsing accuracy on IID, compositional (Comp) splits, and overall test set.

by indicating whether they would be high or low. However, we expect users to have linguistic variations in their queries. To generalize the process of parsing the NL feedback to actionable feedback for all input tokens, we treat it as a sequence labeling task. Specifically, we build a parser that encodes the NL feedback, the bias variable (e.g., gender), and the original task input and produces a sequence of High / Low / NA labels for the complete input token sequence. An example feedback and its parse are shown in Table 1. Such an approach allows us to encode complex feedback on multiple input tokens (see Figure 1).

Since we do not have large annotated data for the parsing task, we instead adopt a few-shot framework, following (Slack et al., 2022). We use a large language model (e.g. GPT-3; text-davinci-003) as they have strong priors for language understanding (here, parsing) tasks from their pre-training phase. We use a few demonstrative parsing examples for in-context learning of the parser. See the parsing task example in Table 1.

### 3.2 Modifying Bias Rationales

After parsing the NL feedback, we use the parse labels to update the bias importance scores. First, we convert each parse label to a numeric equivalent using the following map (parse label $\rightarrow$ important score): High $\rightarrow$ 1; Low $\rightarrow$ 0; NA $\rightarrow$ unchanged. Then we use a linear combination to update the bias importance scores:

$$\text{bias}_{\text{new}} = \alpha \text{bias}_{\text{new}} + (1 - \alpha)\text{bias}_{\text{user}}$$

with $\alpha$ hyperparameter and $\text{bias}_{\text{user}}$ being the numeric equivalent of the user feedback.

### 3.3 Modifying Task Rationales and Prediction

Change in bias importance scores should propagate to the task rationale. We explored two strategies to update the task rationale.

- **Heuristic:** Following (He et al., 2022), we penalize current task importance for a token only

if its updated bias importance is higher than a threshold. The new task rationales are used to generate the new prediction.

- **Gradient:** Since changes in bias rationale scores affect task rationales scores (hence the task rationales), we can directly perturb the final hidden states $h$ of the classification model that generate the task rationale scores for each token (Majumder et al., 2021). We compute a KL-divergence ($\mathcal{K}$) score between $\text{bias}_{\text{old}}$ and $\text{bias}_{\text{new}}$ and and compute its gradient $\nabla_h \mathcal{K}$ w.r.to $h$. Finally, we update $h$ by minimizing the $\mathcal{K}$ via back-propagation using the computed gradients. Note no model parameters are updated in this process. The updated $h$ generates the new task rationales and a new prediction.

## 4 Experiments and Results

We break our experiments into two parts: 1) developing the NL parser and 2) interactive debiasing with INTERFAIR. We use BiosBias (De-Arteaga et al., 2019), a dataset made from a large-scale user study of gender in various occupations. It contains short biographies labeled with gender and profession information, and a possible confluence exists between gender and annotated profession labels.

Using INTERFAIR, we would like to predict the profession from biographies without the influence of gender. Following (Ravfogel et al., 2020), we use 393,423 biographies with binary gender labels (male/female) and 28 professions labels (e.g. professor, model, etc.). We initially used 255,710 examples for training and 39,369 for validation. We use 500 examples (a random sample from the rest 25%) as a test set for interactive debiasing.

For evaluation, we use accuracy for task performance (profession prediction) and use an off-the-shelf gender detector to measure the bias in the task rationales (Bias F1), following He et al. (2022).

## 4.1 NL Feedback Parsing

Following Slack et al. (2022), we use 5, 10, or 20 examples annotated by two independent annotators for the NL parser. We additionally obtain a set of 50 more annotations for testing the parser. While testing the performance of the parser, we use the accuracy metric, i.e., if the parsed feedback matches with the gold parse. We also consider two splits for testing: an IID split where the gold parse contains non-NA labels for one or two contiguous input token sequences and a compositional split where the gold parse has three or more contiguous token sequences. Table 1 shows the parsing accuracy, which reveals that the compositional split is harder than the IID due to its complexity. However, the few-shot parsing using LLMs is faster and easier to adapt with newer user feedback instead of finetuning a supervised model (Slack et al., 2022).

## 4.2 Interactive debiasing

We perform a user study with 10 subjects who interact with INTERFAIR and optionally provide feedback to one of the two objectives – 1) **Constrained:** Minimize bias in task rationales without changing the task prediction, and 2) **Unconstrained:** Minimize bias task rationales as a priority, however, can update task prediction if it seems wrong. The cohort was English-speaking and had an awareness of gender biases but did not have formal education in NLP/ML. The study included an initial training session with 10 instances from the BiosBias test set. Subsequently, participants engaged with 500 reserved examples designated for the interactive debiasing phase. The gender split of the subject pool was 1:1.

To understand the change in model performance and bias, we consider two other debiasing models along with the base model (He et al., 2022) used in INTERFAIR: (1) *Rerank*, an inference-time debiasing variant where the task rationale is considered based on ascending order of bias energy (He et al., 2022); (2) *Adv*, a model trained with an adversarial objective (Zhang et al., 2018) to debias the model's latent space, but incapable of producing any rationales.

Table 2 shows that when we use Full Text as task input, the bias in task rationales is very high. Reranking decreases the bias but also incurs a drop in task performance. The adversarial method does not produce any explanation and cannot use any additional feedback, leading to low task performance.

| Models | Acc. | Bias F1 | Compre. | Suff. |
|---|---|---|---|---|
| Full Text | 81.1 | 0.98 | – | – |
| Reranking | 70.3 | 0.45 | 0.23 | 0.32 |
| Adv | 36.7 | 0.35 | – | – |
| INTERFAIR-base | 80.1 | 0.38 | 0.52 | 0.01 |
| **Constrained:** | | | | |
| INTERFAIR-Heuristic | 80.1 | 0.33 | 0.51 | 0.01 |
| INTERFAIR-Gradient | 80.1 | **0.30** | **0.48** | **0.00** |
| **Unconstrained:** | | | | |
| INTERFAIR-Heuristic | 83.9 | 0.38 | 0.51 | **0.00** |
| INTERFAIR-Gradient | **85.2** | 0.33 | **0.48** | **0.00** |

Table 2: **Evaluation** for task accuracy (Acc. (%) ↑), bias (F1 ↓), and faithfulness for task rationales: Comprehensiveness (Compre. ↑) and Sufficiency (Suff. ↓)

INTERFAIR without feedback balances the task performance and bias very well.

In the constrained setup, the user locks in the task performance (by design) but are able to decrease bias further at the inference time just by perturbing model hidden states using NL feedback. In the unconstrained setup, users are able to modify bias rationales in such a way that improves task performance while decreasing bias. Most importantly, even though 81% (Full Text performance) is the upper bound of accuracy for purely training-based frameworks, users achieve a better task performance (4-5%) while keeping the bias in rationales minimal. In both setups, gradient-based changes in model states are superior to the heuristic strategy to modify the final task rationales. Since unconstrained setup can also confuse users and may lead to failure modes, we see the lowest bias F1 is achieved in the unconstrained setup; however, users were able to keep the bias as low as the INTERFAIR-base model in all interactive settings.

Test-time improvement of task performance and bias with a frozen model indicates that 1) full-text-based training suffers from spurious correlation or noise that hampers task performance, and 2) interactive debiasing is superior to no feedback since it produces better quality human feedback to refine task performance while eliminating bias. This phenomenon can be seen as a proxy for data augmentation leading to a superior disentanglement of original task performance and bias.

Finally, since test-time interactions modify task rationales, we check their faithfulness using comprehensiveness and sufficiency scores, measured as defined in (DeYoung et al., 2020). Sufficiency is defined as the degree to which a rationale is adequate for making a prediction, while comprehensiveness indicates whether all rationales selected

are necessary for making a prediction. A higher comprehensiveness score and a lower sufficiency indicate a high degree of faithfulness. We show that even after modification through interactions, the faithfulness metrics do not deviate significantly from the base models, and final task rationales from INTERFAIR remain faithful.

### 4.3 Discussion

**Feedback format**    In our initial pilot study with a sample size of N=5 (subjects with no background in NLP/ML), we investigated two feedback formats: 1) allowing participants to perturb weights through three options - NA/High/Low, and 2) soliciting natural language feedback. While it may seem more efficient to offer feedback by engaging with individual tokens and selecting a perturbation option, participants expressed confusion regarding how altering the significance of each token would effectively mitigate bias. Conversely, participants found it more intuitive to provide natural language feedback such as "A person's name is unrelated to their profession." To understand the possibility of this would change had our participants possessed a background in NLP/ML, we conducted a supplementary study involving another cohort of 5 participants, all of whom had completed at least one relevant course in NLP/ML. These participants encountered no difficulties in directly manipulating token importance using the NA/High/Low options and revealed a comparable trend to approaches employing natural language feedback methods.

**Beyond LSTMs**    LSTM-based base models enjoyed the gradient update during the interactive debiasing, but to extend this to the model to no hidden states access (e.g., GPT-3), we have to restrict only to heuristic-based approach. We investigate a modular pipeline that uses GPT-3 (`text-davinci-003`) to extract both the task and bias rationales and then followed by an LSTM-based predictor that predicts the task labels only using the task rationales. The rationale extractor and task predictor are not connected parametrically, another reason why we can only use heuristic-based methods to update the task rationales. The final accuracy and Bias F1 were not significantly different than what was achieved in our LSTM-based setup despite GPT-3 based IN-TERFAIR-base having significantly better performance (acc. 84.0). This suggests the choice of the underlying base model may not be significant if the output can be fixed through iterative debiasing.

## 5    Conclusion

In summary, INTERFAIR shows the possibility of user-centric systems where users can improve model performances by interacting with it at the test time. Test-time user feedback can yield better disentanglement than what is achieved algorithmically during training. Debiasing is a subjective task, and users can take the higher agency to guide model predictions without affecting model parameters. However, INTERFAIR does not memorize previous feedback at a loss of generalization, which can be addressed via memory-based interactions (Tandon et al., 2022), or persistent model editing (Mitchell et al., 2021) as future work.

**Acknowledgements**    We thank the anonymous reviewers and the members of the Aristo team at AI2 for their insightful feedback. BPM was funded, in part, by an Adobe Research Fellowship.

## Limitations

Our framework does not persist user feedback which may make the debiasing process repetitive and tedious. Users may have to provide almost identical feedback on different data points where the model is making a systemic error. It should be prudent to store user feedback and apply it automatically and efficiently to minimize the user-in-the-loop effort. We also acknowledge that there can be multiple ways of debiasing a task, and it depends on the context of each example. Also, debiasing being a subjective task at the end, its evaluation rests on the subjective evaluation of the experiments performed. We tried our best to make the subject sample as representative as possible; however, the sample can still suffer from socio-cultural bias.

## Ethics Statement

Our framework assumes that users will not provide any adversarial feedback. We monitored user behavior during the user study and discarded any such feedback from the final evaluation of the system. However, in real-world environments, this assumption may not hold as users can direct the model to generate a more biased prediction than its base performance. However, since we do not have persistent user changes, an adversarial user cannot make a negative impact on another user's session. Still, it is prudent to have monitoring agencies restrict users from directing models to generate biased harmful content.

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
