# OpenReview forum: "InterFair: Debiasing with Natural Language Feedback for Fair Interpretable Predictions"
_EMNLP/2023/Conference — EMNLP 2023 Main_

### Official Review · Reviewer_mchw · 2023-08-04

**Soundness:** 3

**Excitement:**

4: Strong: This paper deepens the understanding of some phenomenon or lowers the barriers to an existing research direction.

**Paper Topic And Main Contributions:**

This paper introduces the InterFair framework for iteratively incorporating user feedback to mitigate the model's reliance on biased features of the input. InterFair proposes first to parse the open-ended user feedback to a form of per-token bias weights with a large language model (GPT-3) used in a few-shot setup. Second, this feedback is used to update the model rationales through differentiation of the model's generated rationales. Results show that users' interactions with model rationales through InterFair can improve the comprehensiveness and sufficiency of models' rationales, without compromising prediction performance on named entity recognition tasks.

**Questions For The Authors:**

A: In the Heuristic approach (L190), how are the new task rationales used to generate the new prediction?

B: In the Gradient approach, you state that you do not update the model, but only the final hidden states (h) of LSTM. My question is how can you differentiate between per-token hidden states of dimensionality (hidden_state_dim, prediction_length) and rationales of (prediction_length, )? Am I missing something here?

**Reasons To Accept:**

* The paper is well-written and intuitively structured
* The motivation of the method's design is clear and well described
* Incorporating human feedback into models' decision process is a potential, though challenging, research direction

**Reasons To Reject:**

My main concern is the underspecification of multiple details throughout the work that are needed to understand the merit of the contribution. Below I list the more important ones, and I follow with assumably easier-to-address points in the "Presentation Improvements" section.

* Background section does not contain an overview of the previous methods addressing debiasing, and specifically debiasing through rationales. This makes it difficult to assess the relatedness and novelty of the proposed approach.
  * Subsequently, the important point of the motivation is that (L92) "even an algorithmically debiased model can have failure modes" probably refers to the previous methods, which, however, remain unknown to the reader at this point.
  * The authors try to gap this bridge in the brief description of their baseline in the Results section, L256-261 (denoted as Rerank and Adv), but I have trouble grasping how these baselines work, not saying how they relate to InterFair, or whether they are sufficiently competitive.

* Evaluation, including baselines, is difficult to interpret. The reader is prompted to understand the main metrics of comprehensiveness and sufficiency only from the cited work (L301). After following this order and checking the ERASER benchmark paper (https://aclanthology.org/2020.acl-main.408.pdf), I am convinced of the selection of these metrics in evaluation, but I believe their thorough description would be in place here.
* I think the current description of the mechanism of incorporating feedback (Sec. 3.3) leaves a lot of space for ambiguity. After multiple reads, I do not feel confident in either the Heuristic nor the Gradient approach description. My questions address this, but the method description, as the main proposition of the paper, should be much more specific. Missing submission of the code or appendices with details currently makes it hardly possible to reproduce and possibly use the method.

**Reproducibility:**

1: Could not reproduce the results here no matter how hard they tried.

**Reviewer Confidence:**

2: Willing to defend my evaluation, but it is fairly likely that I missed some details, didn't understand some central points, or can't be sure about the novelty of the work.

**Typos Grammar Style And Presentation Improvements:**

* I do not understand the description of "compositional split" on L238: "compositional split 238 where the gold parse has three or more contiguous 239 token sequences". Subsequently, I can not relate to the conclusion on L240: "the compositional split is harder than the IID due to its complexity".
* I believe the part on L084 would deserve further justification: "He et al. (2022) argue that weighing less on high bias and high-task important tokens and promoting their low-bias replacements can simultaneously address both of the failure modes.". How can weighting less on high-task important tokens address model failures?
* L202: It seems to be arguable that hidden states (h) are not a part of the model, since you also state that they are used in prediction.
* L256: perhaps missing "(1)"?

---

> ### Author Rebuttal · Authors · 2023-08-29
>
> Thank you for taking the time to review our paper! We were happy to read that you appreciated our work by showing positive enthusiasm about our (novel) research direction and stating that our framework is well-motivated and the experiments sufficiently support our claims.
>
> We acknowledge your clarification questions with more details and believe that we addressed them all. Please see our response below. We would love to address additional questions during the discussion period if anything remains unclear.
>
> As a short paper, we originally aimed to keep our main paper to-the-point, but we would be happy to add the additional details, as requested by you, in the final version. We will also release the codes of the models with the final version for reproducibility.
>
> -----
> **Background for the algorithmically debiased models and details about baselines**
>
> *Thank you for the point. We will add the details in the final version of the main paper to improve readability.*
>
> Details: L92 particularly refers to our no-feedback base model, as obtained from He et al. 2022, and indicates a failure mode, as depicted in Figure 1. This was in the context of L81-83 where we mentioned two possible failure modes for models that debias through rationales: 1) it produces a correct task prediction but with a highly biased rationale, and 2) it produces a wrong task prediction but a rationale with low bias.
>
> [He et al., 2022](https://aclanthology.org/2022.findings-emnlp.431.pdf) is the first attempt at debiasing through rationales, which we also consider as our no-feedback base model. Rerank is an ablation baseline from He et al. 2022, where the task rationale is selected based on a reversed order of bias importance (i.e., a less biased token with a high task energy will be selected). This is equivalent to when a user greedily selects all the tokens that have low bias importance. Hence, it is a competing baseline.
>
> Adv. refers to debiasing methods (such as [Zhang et al., 2018](https://arxiv.org/pdf/1801.07593.pdf) and [Ravfogel et al., 2020](https://aclanthology.org/2020.acl-main.647.pdf)) using adversarial training that debiases the latent space by projecting the hidden representations into the subspace where the bias is no longer predictable. This is one of the most common algorithmically debiased model architectures, hence a competing baseline. However, this model does not produce task rationales and hence cannot be improved through interactions. By comparing with Adv., we investigate if iteratively improving the rationale-based debiasing model’s output is better than that from an alternative black-box debiasing model.
>
> -----------
> **Details about the evaluation: faithfulness metrics**
>
> *Thank you for pointing this out. We will add more details in the main paper.*
>
> Details: Since InferFair further changes the task rationales, it is prudent to make sure that the task rationales remain trustworthy towards the final prediction. Following the [ERASER benchmark ](https://arxiv.org/pdf/1911.03429.pdf)paper, we measure the faithfulness of extracted rationales using two scores: Sufficiency and Comprehensiveness. Sufficiency is defined as the degree to which a rationale is adequate for making a prediction, while comprehensiveness indicates whether all rationales selected are necessary for making a prediction. A higher comprehensiveness score and a lower sufficiency indicate a high degree of faithfulness. We show that even after modification through interactions, the faithfulness metrics do not deviate significantly from the base models, and final task rationales from InterFair remain faithful.
>
> ----
> **Details about the mechanism for the incorporation of feedback**
>
> *More details and justifications will be added in the main paper. For a better understanding of InterFair’s working, we provide a brief pseudocode here.*
>
> ```
> # Pseudocode for InterFair
>
> Initialize:
> done = 0
> Base model generates output, task rationales (+ scores), and bias rationales (+ scores)
> for each i in 1:N:
> 	Feedback, done <- user (to update the bias rationales)
> 	if done:
>             break
>             end if
> 	Updated bias rationales <- feedback, old bias rationales (via linear transformation)
> 	Updated task rationales <- updated bias rationales (via heuristic/gradient method)
> 	Updated prediction, (bias rationales) <- updated task rationales
> end for
> ```
>
> ----
> **How are task rationales used to generate prediction in the heuristic method?**
>
> The rationale selection score (i.e., importance score) is updated using heuristics and then used as soft binary decisions for each token to be selected in the task rationales or not. The soft selection (new task rationales) is then used for prediction.
>
> ----------
> **How do we update the final hidden states in the gradient method?**
>
> A single hidden state, responsible for generating the soft selections of tokens to be in the rationales, is updated using the gradient method. First, we compute the KL divergence, $\mathcal{K}$, between the old and updated bias importance scores (L198-199). $\mathcal{K}$ is a function of the hidden state that generated the bias importance scores, which is a function of the input encoder hidden states of LSTM. The same input encoder hidden states also produce the hidden state accountable for generating the soft selections of tokens to be in the rationales, $h$, as mentioned in L200, which makes  $\mathcal{K}$ a function of $h$. Hence, we can update $h$ through back-propagation by computing the gradient $\nabla_h \mathcal{K}(h)$.
>
> Similar inference-time gradient-based updates have been executed in [Dathathri et al., 2020](https://arxiv.org/pdf/1912.02164.pdf), [Qin et al., 2021](https://arxiv.org/pdf/2010.05906.pdf).
>
> ------
> **Compositional split for the parser data**
>
> Compositional split contains examples that are out-of-distribution from the few-shot examples for our GPT-3 based parser. The in-context examples for GPT-3 parser contained examples whose output had a maximum of two contiguous High or Low labels. However, in the test examples, there were parsing examples (compositional split) that contained more than two contiguous High or Low labels in the output. So the parser has to generalize in these out-of-distribution test examples, and hence we observe a lower performance as compared to when test examples were in-distribution.
>
> ------
> **Justification of rationales selection in He et al., 2022**
>
> L81-83 mention two possible failure modes for models in He et al., 2022: 1) it produces a correct task prediction but with a highly biased rationale, and 2) it produces a wrong task prediction but a rationale with low bias. He et al., 2022 counter this by performing a trade-off, they deliberately drop tokens that have both high-bias and high-task importance by replacing them with low-bias but moderate-task important tokens (a bias-performance trade-off), anticipating that the moderate-task important can also generate a correct prediction.
>
> -----
> **Are hidden states part of the model?**
>
> The hidden state ($h$), as mentioned in L195, is generated as an intermediate outcome during the forward pass of the model, which finally influences the final prediction. However, they are not the parameters of the model. InterFair works by updating the hidden states at the inference time but does not update any parameters of the model.
>
> **Presentation suggestions for L256**
>
> Thank you for noting this. We will fix it in the final version.

---

### Official Review · Reviewer_Kj4w · 2023-08-05

**Soundness:** 4

**Excitement:**

3: Ambivalent: It has merits (e.g., it reports state-of-the-art results, the idea is nice), but there are key weaknesses (e.g., it describes incremental work), and it can significantly benefit from another round of revision. However, I won't object to accepting it if my co-reviewers champion it.

**Paper Topic And Main Contributions:**

This work proposes an approach to better tradeoff between model debiasing and performance via user interaction on task and bias rationales. Experimental results show that the proposed methods can better mitigate model bias as well as achieving better task performance.

**Questions For The Authors:**

In line 121, the author mentioned that “the opaqueness of these models hinders faithful perturbation of reationales”, I wonder why LLms hinders perturbation of rational?

**Reasons To Accept:**

Clear motivation with a good showcase of how this work has done to better tradeoff between model performance and bias.

**Reasons To Reject:**

Main experiments are conducted on LSTM, but have no idea on how the proposed methods will perform on LLMs, which in my view is more worthy of studying. Meanwhile, this work if quite similar to He et al. 2022, but the author doesn’t clearly explain the major shortcoming of He et al. 2022 that this work can better handle.

**Reproducibility:**

4: Could mostly reproduce the results, but there may be some variation because of sample variance or minor variations in their interpretation of the protocol or method.

**Reviewer Confidence:**

4: Quite sure. I tried to check the important points carefully. It's unlikely, though conceivable, that I missed something that should affect my ratings.

---

> ### Author Rebuttal · Authors · 2023-08-29
>
> Thank you for taking the time to review our paper! We were happy to read that you appreciated the work by stating that our framework is aptly motivated and showcases clear evidence of a decrease in bias and improvement in model performance.
>
> We think that all your questions are addressable within the rebuttal period. Please see our response below. We would love to address additional questions during the discussion period if anything is unclear.
>
> -----------
> **Proposed method with LLM as a base model**
>
> *Similar trends are observed when InterFair uses LLMs as base models as compared to a no-feedback baseline. We will add additional results in the final version.*
>
> Details: As a short paper, we initially aimed to keep our results self-contained, and hence we only used the LSTM model for the BiosBias dataset, extending He et al., 2022. However, our method can be applied to any base model (e.g., BERT) whose hidden states and gradients can be accessed.
>
> To support our claim, we run two additional sets of experiments, with similar human study setup as the main paper:
>
> 1) a BERT-based model with two separate heads for task and bias rationales, and a final layer that predicts the task prediction, only using selected task rationales as bottleneck. Since both task and bias rationale generators are integrated with the final predictor, we can use both heuristics and gradient-based methods to iteratively debias the outputs further.
>
> | Models | Acc. $\uparrow$ | Bias F1 $\downarrow$ | Compre. $\uparrow$ | Suff. $\downarrow$ |
> |-----|-----|-----|-----|-----|
> |InterFair Base| 86.2 | 0.37 | 0.48 | 0.01 |
> |Constrained|||||
> |InterFair Heuristic| 86.2 | 0.32 ($\downarrow$0.05) | 0.50 | 0.01 |
> |InterFair Gradient| 86.2 | **0.29** ($\downarrow$0.08) | 0.47 | 0.01|
> |Unconstrained|||||
> |InterFair Heuristic| 87.4 ($\uparrow$1.2) | 0.37 (0.0) | 0.48 | 0.00|
> |InterFair Gradient| **87.8** ($\uparrow$1.6) | 0.32 ($\downarrow$0.05) | 0.48 | 0.00|
>
> (*numbers in parenthesis measure gains/losses from InterFair Base entries in the table)
>
>
> 2) a modular pipeline that uses GPT-3 (text-davinci-003) to extract both the task and bias rationales and then followed by an LSTM-based predictor that predicts the task labels only using the task rationales. The rationale extractor and task predictor are not connected parametrically; hence we can only use heuristic-based methods to update the task rationales.
>
> | Models | Acc. $\uparrow$ | Bias F1 $\downarrow$ | Compre. $\uparrow$ | Suff. $\downarrow$ |
> |-----|-----|-----|-----|-----|
> |InterFair Base| 84.0 | 0.23 | 0.30 | 0.00 |
> |Constrained|||||
> |InterFair Heuristic| 84.0 | **0.21** ($\downarrow$0.02) | 0.30 | 0.00 |
> |Unconstrained|||||
> |InterFair Heuristic| **84.8** ($\uparrow$0.8) | 0.23 (0.0) | 0.31 | 0.01|
>
> (*numbers in parenthesis measure gains/losses from InterFair Base entries in the table)
>
> We will add the additional results in the final version, as we concur with you that adding these would increase the accessibility of our work.
>
> -----------
> **Difference from He et al. 2022**
>
> *He et al. 2022 is our no-feedback baseline that does not include interactive updation. Our proposed methods outperform the baseline.*
>
> Details: The key difference between He et al., 2022 and ours lies in the motivation. He et al., 2022 debias their models by a fixed definition: tokens that exhibit both a high bias and high task importance are discarded algorithmically. However, this solution may not be perfect due to the limit of learning statistical patterns from the data or in cases where the bias-performance trade-off is a subjective decision specific to an end-user. To achieve this, we propose InterFair, which allows interactive control (a key technical difference from He et al., 2022) over the task rationales by perturbing bias encoded in the model parameters, which shows superior performance in terms of both decreasing bias and improving task performance.
>
> ------------
> **Opaqueness in LLM-based rationales**
>
> In L121, our comment was a justification of not using black-box LLMs (GPT-3, GPT-3.5) since we do not have access to the hidden states in these LLMs responsible for generating extractive rationales leading to opaqueness, while we can clinically attribute to the responsible states for rationale generation in the LSTM architecture used in He et al., 2022 or BERT-based model we reported additional results on.

---

### Official Review · Reviewer_beK8 · 2023-08-12

**Soundness:** 4

**Excitement:**

3: Ambivalent: It has merits (e.g., it reports state-of-the-art results, the idea is nice), but there are key weaknesses (e.g., it describes incremental work), and it can significantly benefit from another round of revision. However, I won't object to accepting it if my co-reviewers champion it.

**Paper Topic And Main Contributions:**

Firstly, the paper proposes a new pipeline (InterFair) to collect human feedbacks about bias rationales and incorporate that feedback into how the model produces future task rationales and bias rationales.
Secondly, the paper does an experiment and a human study to test their proposed pipeline.

**Questions For The Authors:**

Question A: Could you please provide some potential explanations on why gradient methods outperform heuristic methods (Table 2)?

**Reasons To Accept:**

1. clear motivation on why human feedback matters
2. novel methodology
3. results show that their proposed method outperforms the no-feedback baseline

**Reasons To Reject:**

1. I find it unclear on why users should provide natural language feedback to be parsed into High/Low/NA importance score per token, if the goal is just to adjust the weights of the already highlighted bias rationales? As I guess changing weights of a few tokens is easier/faster than writing a free-text response (to be later parsed by a GPT-3 LLM) anyway, why don't the authors only keep the option: ``To directly modify the bias rationales, users can increase or decrease the bias importance score for each token accordingly.'' Without more detailed motivation, the choice of natural language feedback may be an unnecessary hurdle for the experiment.
2. Lacking details of human study (e.g. platform, subjects' background in NLP/ML bias, the distribution of feedback style in natural language form vs. in importance weight adjustment form)

**Reproducibility:**

3: Could reproduce the results with some difficulty. The settings of parameters are underspecified or subjectively determined; the training/evaluation data are not widely available.

**Reviewer Confidence:**

3: Pretty sure, but there's a chance I missed something. Although I have a good feel for this area in general, I did not carefully check the paper's details, e.g., the math, experimental design, or novelty.

---

> ### Author Rebuttal · Authors · 2023-08-29
>
> Thank you for taking the time to review our paper! We were happy to read that you appreciated the work by stating that our framework for incorporating human feedback is 1) novel, 2) clearly motivated, and 3) showcases superior results as compared to competing baselines that strongly support our claims.
>
> We addressed all of your questions with more details and additional experiments. Please see our response below. We would love to address additional questions during the discussion period if anything is unclear.
>
> -----------
> **Justification for using Natural Language Feedback**
>
> *Our pilot study revealed that users who do not have a background in NLP/ML found it easier to provide natural language feedback as opposed to changing the importance scores directly, which required some understanding of the underlying model.*
>
> Details: In our initial pilot study with a sample size of N=5 (subjects with no background in NLP/ML), we investigated two feedback formats: 1) allowing participants to perturb weights through three options - NA/High/Low, and 2) soliciting natural language feedback. While it may seem more efficient to offer feedback by engaging with individual tokens and selecting a perturbation option, participants expressed confusion regarding how altering the significance of each token would effectively mitigate bias. Conversely, participants found it more intuitive to provide natural language feedback such as "A person's name is unrelated to their profession." Notably, this situation might have evolved had our participants possessed a background in NLP/ML.
>
> To corroborate this, we conducted a supplementary study involving another cohort of 5 participants, all of whom had completed at least one relevant course in NLP/ML. These participants encountered no difficulties in directly manipulating token importance using the NA/High/Low options. Despite the reasonable accuracy of our GPT-3 parser, directly adjusting token importance offers distinct advantages, i.e., zero parsing errors. Our findings reveal a comparable trend to approaches employing natural language feedback methods.
>
> | Models | Acc. $\uparrow$ | Bias F1 $\downarrow$ | Compre. $\uparrow$ | Suff. $\downarrow$ |
> |-|-|-|-|-|
> |Constrained|||||
> |InterFair Heuristic| 80.1 | 0.32 ($\downarrow$0.02) | 0.50 | 0.01 |
> |InterFair Gradient| 80.1 | **0.29** ($\downarrow$0.02) | 0.47 | 0.01|
> |Unconstrained|||||
> |InterFair Heuristic| 84.2 ($\uparrow$0.3) | 0.37 ($\downarrow$0.01) | 0.50 | 0.00|
> |InterFair Gradient| **85.6** ($\uparrow$0.4) | 0.32 ($\downarrow$0.01) | 0.47 | 0.00|
>
> (*numbers in parenthesis measure gains/losses from respective entries in Table 2, where all feedback was in natural language)
>
> ---------
> **Q: Details about the human study**
>
> *Thank you for mentioning this. We will add more details in the final version.*
>
> Details: In our human study (from the main paper), a cohort of 10 English-speaking individuals was selected. These participants had an awareness of gender biases but did not have formal education in NLP/ML. The study included an initial training session with 10 instances from the BiosBias test set. Subsequently, participants engaged with 500 reserved examples designated for the interactive debiasing phase. The gender split of the subject pool was 1:1.
>
> In the newer study (with token-level importance perturbation), the setup was similar, but the subjects had a formal foundation in NLP/ML. Of these participants, 3 out of 5 were female, and the rest were male.
>
> We did not collect both forms of feedback in a single study; however, our pilot (with inexperienced subjects in NLP/ML) revealed only 1 out of 5 subjects were comfortable with furnishing token-level feedback via direct manipulation of importance scores.
>
> -------------
> **Why is the gradient method better than the heuristic method**
>
> *The gradient method changes the task rationales through the learned non-linear activations in the base model, which is more accurate and better calibrated than the linear changes made by the heuristic methods.*
>
> Details: As mentioned in Section 3.2, we follow He et al., 2022, wherein the task importance is linearly reduced by the quantity (updated bias importance for the token - A) if the difference is positive; otherwise, the task importance score remains unchanged. Here, A, the threshold, is a hyperparameter. However, a linear adjustment (a drawback in He et al., 2022) in task importance score may not necessarily yield the desired change in the outcome.
>
> To accurately calibrate the updation of the task rationales, we employ gradient methods to modify the singular hidden state accountable for generating the task importance scores for tokens. This modification is based on minimizing the KL divergence between the previous and updated bias importance distributions. This approach is feasible due to the construction of both hidden vectors responsible for task and bias importance scores from the same encoder hidden states.
>
> By definition, the gradient method produces smoother updates to task rationales based on user feedback, possibly leading to superior performance compared to heuristic methods.

---

### Meta-Review · Area_Chair_QLJR · 2023-09-20

**Recommendation:** 4

**Metareview:**

The authors propose InterFair, a novel human-in-the-loop pipeline approach to model debiasing, where free-text feedback from humans is used to update importance scores for model rationales. The free-text feedback is parsed by a text-davinci-003 model and transformed into numeric scores, which are then used to update the bias importance of each individual token for a frozen model on the BiosBias classification task. Following the claim that it is challenging to fairly eliminate biases algorithmically, the authors evaluate their scenario at inference-time for two experiments, where users interact with test samples and inputs, showing that both scenarios further mitigate bias from explanations.

The reviewers agree that the paper is clearly motivated (beK8, KjGw, mchw), well written and inuitively structured (mchw). Reviewers note that the proposed methodology is novel (beK8) and has a lot of potential (mchw) — and that the presented experimental results support the authors’ approach (beK8, Kj4w).

The main criticisms presented in the reviews are lack of details regarding the human study (beK8) and more clarity with respect to the gap to previous work (beK8, Kj4w) as well as the experimental setup (mchw). Furthermore, the choice of a LSTM-based model as part of the pipeline and not a Transformer-based variant limits the scope of the work (Kj4w). Lastly, a question was raised with respect to the choice of natural language feedback instead of using plain token editing (beK8).

The authors responded to the questions raised in the reviews, committing to add clarifying details. The authors also presented additional results with a BERT-based model, motivated their choice of free-text input through a user study and clarified the gap with respect to previous work.

Upon reading the paper, reviews, as well as the discussion, it is my opinion that the paper is for the most part well written and easy to follow. Furthermore, it tackles a relevant problem in a novel manner. When reading the paper, my main concerns come from a significant reliance on work of He et al, where the reader is assumed to be familiar with the setup — something that can be improved in the manuscript. Another concern was the limited scope of experiments, which is addressed by the authors in the discussion period by adding experimental results on a Transformer-based model.

---

### Decision · Program_Chairs · 2023-10-07

**Decision:**

Accept-Main

**Comment:**

The authors propose InterFair, a novel human-in-the-loop pipeline approach to model debiasing, where free-text feedback from humans is used to update importance scores for model rationales. The free-text feedback is parsed by a text-davinci-003 model and transformed into numeric scores, which are then used to update the bias importance of each individual token for a frozen model on the BiosBias classification task. Following the claim that it is challenging to fairly eliminate biases algorithmically, the authors evaluate their scenario at inference-time for two experiments, where users interact with test samples and inputs, showing that both scenarios further mitigate bias from explanations.

The reviewers agree that the paper is clearly motivated (beK8, KjGw, mchw), well written and inuitively structured (mchw). Reviewers note that the proposed methodology is novel (beK8) and has a lot of potential (mchw) — and that the presented experimental results support the authors’ approach (beK8, Kj4w).

The main criticisms presented in the reviews are lack of details regarding the human study (beK8) and more clarity with respect to the gap to previous work (beK8, Kj4w) as well as the experimental setup (mchw). Furthermore, the choice of a LSTM-based model as part of the pipeline and not a Transformer-based variant limits the scope of the work (Kj4w). Lastly, a question was raised with respect to the choice of natural language feedback instead of using plain token editing (beK8).

The authors responded to the questions raised in the reviews, committing to add clarifying details. The authors also presented additional results with a BERT-based model, motivated their choice of free-text input through a user study and clarified the gap with respect to previous work.

Upon reading the paper, reviews, as well as the discussion, it is my opinion that the paper is for the most part well written and easy to follow. Furthermore, it tackles a relevant problem in a novel manner. When reading the paper, my main concerns come from a significant reliance on work of He et al, where the reader is assumed to be familiar with the setup — something that can be improved in the manuscript. Another concern was the limited scope of experiments, which is addressed by the authors in the discussion period by adding experimental results on a Transformer-based model.